# Association between diabetes mellitus and impaired single-leg stance in patients with chronic liver disease: A cross-sectional study

Kenichi Fudeyasu[1,2], Makoto Asaeda[1,3*], Toshihiro Kawae[4], Takuo Nomura[5], Yuki Nakashima[1], Daisuke Iwaki[1], Kouki Fukuhara[1], Takeya Araki[3], Haruya Ohno[6], Eisuke Murakami[7], Shiro Oka[7], Yukio Mikami[3]

1 Division of Rehabilitation, Department of Clinical Practice and Support, Hiroshima University Hospital, Hiroshima, Japan, 2 Department of Rehabilitation, Osaka Medical and Pharmaceutical University Hospital, Osaka, Japan, 3 Department of Rehabilitation Medicine, Hiroshima University Hospital, Hiroshima, Japan, 4 Department of Physical Therapy, Makuhari Human Care Faculty, Tohto University, Chiba, Japan, 5 Department of Physical Therapy, Faculty of Rehabilitation, Kansai Medical University, Osaka, Japan, 6 Endocrinology and Diabetic Medicine, Hiroshima University Hospital, Hiroshima, Japan, 7 Department of Gastroenterology, Hiroshima University Hospital, Hiroshima, Japan

* asaedam@hiroshima-u.ac.jp

## Abstract

Diabetes mellitus (DM) is highly prevalent among patients with chronic liver disease (CLD) and is associated with disease progression and complications. However, the impact of DM on physical function, particularly balance, in patients with CLD remains unclear. The aim of this study was to investigate the association between DM and physical function in patients with CLD, with a specific focus on impaired single-leg stance test (SLST). This retrospective study analyzed the medical records of patients with CLD at Hiroshima University Hospital between 2014 and 2017. Logistic regression analysis was performed to identify factors associated with impaired SLST. Receiver operating characteristic curve analysis was conducted to determine cutoff values for predictive factors. The analysis included 152 patients with CLD, of whom 78% had comorbid DM. Patients with DM had a significantly higher prevalence of impaired SLST than those without DM (20% vs. 0%, p = 0.002, Cramer's $V$ 0.23). In 118 patients with CLD who had comorbid DM, age (odds ratio [OR] 1.089, 95% confidence interval [CI] 1.020–1.177, p = 0.009), body mass index (OR 1.176, 95% CI 1.045–1.343, p = 0.006), and extracellular water-to-total body water ratio (ECW/TBW) (OR 1.065, 95% CI 1.003–1.138, p = 0.039) were significant independent factors associated with impaired SLST (Nagelkerke pseudo-$R^2$ 0.31, p < 0.001). The ECW/TBW had the highest predictive accuracy, with a cutoff value of 0.393 (area under the curve = 0.733, sensitivity = 73.9%, specificity = 68.8%). DM was associated with impaired SLST in patients with CLD, suggesting a decline in balance. Age, body mass index, and ECW/TBW are significant predictors of impaired SLST. An ECW/TBW ratio of 0.393 indicates

**Data availability statement:** All relevant data underlying the results of this study are available in the Hiroshima University Institutional Repository, accessible at https://hiroshima.repo.nii.ac.jp/records/2040729 (DOI: 10.15027/0002040729).

**Funding:** This work was supported by a grant (H30-A59) from the Japanese Society of Physical Therapy and Japan Society for the Promotion of Science KAKENHI Grant Number JP19K19831. "The funders had no role in study design, data collection and analysis, decision to publish, or preparation of the manuscript".

**Competing interests:** The authors declare that they have no competing interests.

"subclinical" edema in patients with CLD and DM and should be considered in the assessment of fall risk.

## Introduction

Chronic liver disease (CLD) affects approximately 1.5 billion people worldwide, and its incidence, along with cirrhosis, has increased by 13% since 2000 [1]. Recent studies have highlighted the high prevalence of diabetes mellitus (DM) among patients with CLD [2,3]. A 2019 meta-analysis reported that 56% of patients with nonalcoholic fatty liver disease had comorbid type 2 DM, indicating that DM is more common in CLD than previously considered [3]. Furthermore, among patients with CLD referred from hepatology to rehabilitation departments, 83% had comorbid DM [4]. Several studies have shown that DM increases the risk of complications in patients with cirrhosis, including hepatic encephalopathy, spontaneous bacterial peritonitis, and variceal bleeding. Moreover, DM is associated with a higher incidence of hepatocellular carcinoma (HCC) [2,5]. In addition to its role in the progression of CLD [5,6], DM is also recognized as a risk factor for mortality in patients with cirrhosis [2,7].

CLD progression increases the occurrence of sarcopenia, with a notable decline in physical function such as grip strength and gait speed [4,8,9]. Cirrhosis is associated with a high annual incidence of falls (approximately 30%), significantly exceeding rates observed in community-dwelling populations [9,10]. Cirrhosis is also associated with increased healthcare costs and greater caregiver burden [11]. DM further exacerbates these issues by increasing sarcopenia prevalence and impairing muscle strength [12,13]. Moreover, DM-related complications, particularly peripheral neuropathy, contribute to impaired balance and an elevated risk of falls [14–16].

Despite the high prevalence of DM among patients with CLD, its impact on physical function remains unclear. We hypothesized that comorbid DM is associated with impaired balance in patients with CLD and that this decline would be evident in the single-leg stance test (SLST) as a clinical measure. Given that DM-related complications are known to impair balance and reduce SLST duration [17–19], the aim of this study was to examine whether comorbid DM is associated with impaired SLST in patients with CLD.

## Materials and methods

### Study design and patients

This was a retrospective cross-sectional study. Data for this study were retrospectively collected from medical records in August 2024. Patients with CLD who were admitted to the Department of Gastroenterology at Hiroshima University Hospital between September 2014 and June 2017 and who were treated in the Department of Rehabilitation Medicine and agreed to receive physical

therapy from a physical therapist were eligible for this study. All patients were hospitalized for CLD, including clinically diagnosed HCC. Exclusion criteria included pacemaker use—as the presence of such made patients unsuitable for body composition assessment—neuromuscular diseases, excessive pain, dementia, and admission for preoperative evaluation for weight loss surgery (Fig 1). The study was conducted in accordance with the Declaration of Helsinki and the Ethical Guidelines for Medical and Health Research Involving Human Subjects enacted by the Ministry of Health, Labour and Welfare of Japan (https://www.mext.go.jp/content/20250325-mxt_life-000035486-01.pdf) and approved by the Ethical Review Committee of Hiroshima University Hospital (Permit No. E-583-1). All patients were verbally informed that their medical records and charts might be used for research purposes. Data were obtained during routine medical care and reviewed retrospectively. The need to obtain informed consent from the study participants was waived by the Ethical Review Committee of Hiroshima University Hospital due to the retrospective nature of the study. Patients who were eligible for this study had the opportunity to refuse to participate in the study by opting out.

Patient characteristics and blood test data were evaluated at the time of hospital admission. Body composition and physical function were assessed at the initiation of physical therapy. The interval between admission and the start of physical therapy was less than three days. Body composition assessment was performed first, prior to the initial physical therapy session. Physical functions, such as SLST and muscle strength, were evaluated by physical therapists in the hospital rehabilitation room. Sufficient rest periods were provided between each test to ensure no interference with the results.

## Clinical and laboratory assessment

Patient age, sex, height, weight, body mass index (BMI), and blood data (hemoglobin [HGB], platelet [PLT] count, total bilirubin [T-BIL], aspartate aminotransferase [AST], alanine aminotransferase [ALT], albumin [ALB], prothrombin time [PT], ammonia [$NH_3$], triglycerides [TG], uric acid [UA], and hemoglobin A1c [HbA1c]) were extracted from medical records. Diagnoses of chronic hepatitis, cirrhosis, hepatic encephalopathy, ascites, HCC, hypertension, dyslipidemia, and DM were also recorded. The Child–Pugh classification and score—assessed by hepatologists—were used to determine liver disease severity, with a score ≥ 7 indicating decompensated cirrhosis.

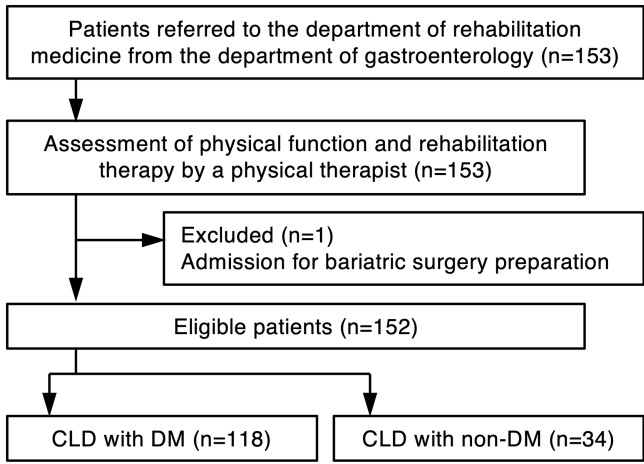

**Fig 1. Flowchart of patient selection for the study.** DM, diabetes mellitus; CLD, chronic liver disease.

## Liver fibrosis assessments

Liver fibrosis was assessed using serum marker-based scoring systems, including the AST-to-platelet ratio index (APRI) and the fibrosis-4 (FIB-4) index [20–24]. The APRI and FIB-4 index were calculated using the following equations,

$$APRI = \frac{AST\left[\frac{IU}{L}\right]/upper\ limit\ of\ normal\ AST\left[\frac{IU}{L}\right]}{PLT\left[\frac{10^9}{L}\right]} \times 100$$

(1)

$$FIB-4\ index = \frac{AST\left[\frac{IU}{L}\right] \times age\ [years]}{PLT\left[\frac{10^9}{L}\right] \times \sqrt{ALT\left[\frac{IU}{L}\right]}}$$

(2)

## Body composition assessment

Skeletal muscle mass was measured using bioelectrical impedance analysis (InBody720; InBody Co., Ltd., Seoul, Korea). The validity of this device for assessing muscle mass has been previously established in patients with CLD [8,25]. To ensure data consistency, all measurements were conducted during the daytime, after voiding, and at least 2 hours after the last meal. While a strict fasting period of at least 8 hours is typically required for high-precision research, a 2-hour post-prandial interval is commonly adopted in clinical settings for practical feasibility [26,27]. This approach is supported by evidence from previous studies showing that, although meal-induced impedance fluctuations can peak within several hours after food intake, these biological changes result in only minor clinical measurement errors for skeletal muscle mass [26]. Additionally, participants were instructed to avoid strenuous exercise and excessive fluid intake prior to the assessment.

Appendicular skeletal muscle mass (ASM) was calculated as the sum of muscle mass in all four limbs. The skeletal muscle mass index (SMI) was derived by dividing ASM by height squared (kg/m²) [8,25,28].

Upper limb muscle mass was calculated by dividing the sum of skeletal muscle mass in both upper limbs by height squared. Additionally, the extracellular water-to-total body water ratio (ECW/TBW) was calculated from the bioelectrical impedance analysis to assess total body water balance.

## SLST

The SLST was performed twice on each leg with the patient's eyes open and both arms hanging naturally at their sides [29]. The duration from when the patient lifted one leg until it touched the ground again was measured using a stopwatch, with a maximum limit of 60 seconds. The SLST is recognized as a reliable method with high inter-rater reliability, particularly when the maximum value from multiple trials is used [30]. Previous studies have reported no significant difference in balance between the dominant and non-dominant legs during the SLST [31]. As our participants did not have any musculoskeletal or neuromuscular disorders, we used the average of the maximum values from each leg for the analysis. To account for potential concerns regarding lateral differences, we also calculated the maximum value for each leg individually.

SLST is widely used as a screening tool for fall risk and locomotive syndrome [32–34]. Although some studies have reported limitations of SLST [35,36], others support its utility as a reliable assessment of fall risk. We adopted an SLST duration of < 5 s as an indicator of increased fall risk, as this threshold has been previously reported to correspond to a sensitivity of 0.33 and a specificity of 0.712 for fall prediction [33].

## Muscle strength assessment

Upper and lower limb muscle strength was evaluated.

• Upper limb strength: Grip strength was assessed based on the 2019 Asian Working Group for Sarcopenia (AWGS) criteria. Measurements were taken in 0.5 kg increments using a Smedley-type grip strength meter (Matsuyoshi Medical

Industry Co., Ltd., Tokyo, Japan). Patients stood with their arms relaxed and elbows extended. The maximum grip strength of two trials for each arm was recorded, and the average of the highest values was used for the analysis.

- Lower extremity muscle strength: Knee extension force (KEF) was measured using a belt-mounted handheld dynamometer (μTas F-1, Anima Co., Tokyo, Japan). Patients were seated with the trunk upright, the knee flexed at 90°, and one lower leg fixed with a belt. Two trials per side were performed, and the highest values were averaged. KEF was normalized to body weight [13,37].

## Gait speed assessment

Patients walked a 16-m path, with acceleration during the initial 3 m and deceleration during the final 3 m. The time to walk the middle 10 m was measured using a stopwatch, and gait speed was calculated [28]. Patients walked at a comfortable pace without assistive devices. The test was repeated twice, and the average value was used.

Impaired gait was defined as a gait speed of < 1.0 m/s, based on criteria established by the 2019 AWGS [28].

## Diagnosis of sarcopenia

The diagnostic criteria for sarcopenia used in this study were those defined by the AWGS 2019 [28] and the Japanese Society of Hepatology (JSH; 2nd edition) [8,25]. According to the AWGS 2019 diagnostic criteria, sarcopenia is defined as a decrease in skeletal muscle mass and a decrease in muscle strength or gait speed. The AWGS 2019 criteria for decreased skeletal muscle mass include an SMI of < 7.0 kg/m² in men and < 5.7 kg/m² in women, and the criteria for decreased muscle strength include a grip strength of < 28 kg in men and < 18 kg in women or a decreased gait speed of < 1.0 m/s [28].

The JSH (2nd edition) diagnostic criteria for decreased skeletal muscle mass and strength are equivalent to sarcopenia. The criteria for decreased skeletal muscle mass in the JSH are defined as an SMI of < 7.0 kg/m² in men and < 5.7 kg/m² in women, and the criteria for muscle weakness are a grip strength of < 28 kg in men and < 18 kg in women [8,25].

## Statistical analysis

All continuous variables were first subjected to the Shapiro–Wilk test to assess normality. Variables that followed a normal distribution were presented as mean (± standard deviation) and were compared between groups using the Student's t-test, with Cohen's $d$ calculated as the effect size. Variables that deviated from normality were expressed as median (interquartile range) and compared using the Mann–Whitney U test. For the U test, the effect size $r$ was calculated using the formula $r = |Z|/\sqrt{N}$. Categorical variables were presented as absolute numbers and percentages and compared using the Fisher's exact test or chi-squared test, as appropriate. The association for categorical variables was evaluated using Cramer's $V$ as the effect size. Effect sizes were interpreted using the following criteria: Cohen's $d$ (0.2–0.5 = small, 0.5–0.8 = medium, and ≥ 0.8 = large), $r$, and Cramer's $V$ (0.1–0.3 = small, 0.3–0.5 = medium, and ≥ 0.5 = large) [38].

An additional analysis was conducted using data from 118 patients with CLD who had comorbid DM to identify factors associated with impaired SLST in this subgroup. Logistic regression analysis was performed with impaired SLST (< 5 s) as the dependent variable, using both univariable and multivariable analysis models. Odds ratios (ORs) with 95% confidence intervals (CIs) were reported, and the model fit was assessed using the Nagelkerke pseudo-$R^2$. Following previous studies, KEF/weight values were multiplied by 100 and ECW/TBW values by 1000 to facilitate interpretation [39]. Multicollinearity was assessed by calculating variance inflation factors, with values ≥ 2 considered indicative of collinearity.

Receiver operating characteristic (ROC) curve analysis was conducted to evaluate the predictive accuracy of factors identified in multivariate analysis. The optimal cutoff value for predicting impaired SLST was determined using the Youden index, which maximizes the sum of sensitivity and specificity (sensitivity + specificity − 1). Sensitivity, specificity, negative predictive value (NPV), and positive predictive value (PPV) were calculated to assess diagnostic performance. ROC

curve analysis and area under the curve (AUC) were also computed, with the following criteria for discrimination: AUC 0.7–0.8 = acceptable, AUC 0.8–0.9 = excellent, and AUC ≥ 0.9 = outstanding [40].

All statistical tests were two-sided, and statistical significance was set at p < 0.05. All statistical analyses were performed using JMP® Pro (version 18.0.1; SAS Institute Inc., Cary, NC, USA). Effect sizes that were not automatically generated by JMP ($r$ and Cramer's $V$) were manually calculated using Microsoft Excel based on the test statistics obtained from JMP.

## Results

### Patient characteristics

The study included 152 patients with CLD, with a median Child–Pugh score of 5 (5–6). Among these patients, 83% were classified as Child–Pugh Class A. The prevalence of DM was 78%. The etiologies of CLD included hepatitis C virus (HCV; 38%), metabolic dysfunction-associated steatotic liver disease (MASLD, 32%), hepatitis B virus (HBV; 12%), alcohol-associated liver disease (ALD; 11%), and other causes (7%) (Tables 1 and S1).

### Comparison of patients with CLD with and without DM

To evaluate the impact of DM on CLD, we compared patient characteristics and clinical parameters between patients with and without DM (Tables 2 and S2). The DM group was significantly older than the non-DM group (68 [63–73] vs. 63 [56–71] years, p = 0.005). The DM group also had a lower BMI (24.7 [22.0–28.8] vs. 27.1 [24.3–30.6] kg/m², p = 0.025) and lower levels of liver enzymes, including AST (29 [22–46] vs. 42 [28–61] IU/L, p = 0.004) and ALT (26 [18–47] vs. 42 [27–75] IU/L, p = 0.001). As expected, HbA1c levels were significantly higher in the DM group (7.2 [6.6–8.0] vs. 5.8 [5.4–6.1] %, p < 0.001).

Regarding physical function, no significant differences were observed in gait speed or sarcopenia prevalence between the two groups. However, the DM group exhibited significantly lower physical performance, with reduced KEF/weight (0.50 ± 0.15 vs. 0.57 ± 0.17 kgf/kg, p = 0.011) and SLST duration (23.9 [7.0–52.8] vs. 49.3 [22.9–60.0] s, p = 0.003). The proportion of patients with impaired SLST (< 5s), defined as the average of the maximum values from each leg, was significantly higher in the DM group (23 [20%] vs. 0 [0%], p = 0.002). Additionally, grip strength was significantly lower in the DM group (25.3 [19.4–32.5] vs. 28.8 [21.9–38.3] kg, p = 0.031) (Table 3).

### Factors associated with impaired SLST

The univariate logistic regression analysis showed that significant factors associated with impaired SLST included age (OR 1.070, 95% CI 1.013–1.139), BMI (OR 1.095, 95% CI 1.007–1.194), ECW/TBW × 1000 (OR 1.077, 95% CI 1.028–1.140), and KEF/weight × 100 (OR 0.951, 95% CI 0.917–0.983) (Table 4). In the multivariate logistic regression analysis, age (OR 1.089, 95% CI 1.020–1.177), BMI (OR 1.176, 95% CI 1.045–1.343), and ECW/TBW × 1000 (OR 1.065, 95% CI 1.003–1.138) remained significant independent factors associated with impaired SLST (Nagelkerke pseudo-$R^2$ 0.31, p < 0.001). Furthermore, VIF values did not exceed 2 for any factor, suggesting the absence of multicollinearity.

### Predictors of impaired SLST

ROC curve analysis was performed to assess the predictive value of variables identified in logistic regression for impaired SLST. The AUC for age was 0.658 (p = 0.014), with a cutoff value of 74 years; sensitivity, 47.8%; specificity, 82.8%; PPV, 40.7%; and NPV, 86.5% (Fig 2a). The AUC for BMI was 0.626 (p = 0.035), with a cutoff value of 27.9 kg/m²; sensitivity, 60.9%; specificity, 76.3%; PPV, 38.9%; and NPV, 88.8% (Fig 2b). The AUC for ECW/TBW was 0.733 (p = 0.001), with a cutoff value of 0.393; sensitivity, 73.9%; specificity, 68.8%; PPV, 37.0%; and NPV, 91.4% (Fig 2c). The AUC for KEF/weight was 0.673 (p = 0.003), with a cutoff value of 0.54 kgf/kg; sensitivity, 82.6%; specificity, 47.3%; PPV, 27.9%; and

**Table 1. Characteristics of the patients included in this study (N = 152).**

| Characteristics | Values |
|---|---|
| Age (years) | 68 (61–73) |
| Sex (female) | 64 (42) |
| Height (cm) | 160.5 (153.9–166.9) |
| Weight (kg) | 63.5 (57.6–75.3) |
| BMI (kg/m²) | 25.1 (22.4–29.4) |
| Blood data | |
| HGB (g/dL) | 13.5 (12.3–14.9) |
| PLT (10⁴/µL) | 15.9 (10.9–21.3) |
| T-BIL (mg/dL) | 0.9 (0.7–1.2) |
| AST (IU/L) | 32 (23–48) |
| ALT (IU/L) | 28 (19–52) |
| ALB (g/dL) | 4.2 (3.7–4.5) |
| PT (%) | 88 (76–97) |
| NH₃ (µmol/L) | 36 (28–49) |
| HbA1c (%) | 6.8 (6.1–7.7) |
| Etiology (ALD/HBV/HCV/MASLD/other) | 17 (11)/ 18 (12)/ 58 (38)/ 49 (32)/ 10 (7) |
| LC | 14 (9) |
| HCC | 32 (9) |
| Encephalopathy | 1 (1) |
| Ascites | 12 (8) |
| Child–Pugh score | 5 (5–6) |
| Uncompensated cirrhosis; Child–Pugh score ≥ 7 | 24 (16) |
| Child–Pugh classification (A/B/C) | 127 (83)/ 18 (12)/ 7 (5) |
| ECW/TBW | 0.39 (0.384–0.395) |
| SMI (kg/m²) | 7.10 (6.40–7.76) |
| Male < 7.0, Female < 5.7 | 41 (27) |
| KEF/weight (kgf/kg) | 0.51 ± 0.16 |
| SLST (s), Mean (Both legs) | 28.3 (9.5–57.4) |
| < 5 | 23 (15) |
| SLST (s), Max (Either leg) | 36.9 (10.9–60) |
| < 5 | 16 (11) |
| Grip strength (kg) | 26.3 (19.9–33.6) |
| Male < 28, Female < 18 | 48 (32) |
| Gait speed (m/s) | 1.12 ± 0.24 |
| < 1.0 | 41 (27) |
| Complications | |
| DM | 118 (78) |
| HT | 80 (53) |
| DL | 38 (25) |
| Sarcopenia by 2019 AWGS | 23 (15) |
| Sarcopenia by JSH, 2ⁿᵈ edition | 21 (14) |

Continuous variables are presented as mean ± standard deviation or median (interquartile range), as appropriate. Categorical variables are presented as numbers (percentages). BMI, body mass index; HGB, hemoglobin; PLT, platelet; T-BIL, total bilirubin; AST, aspartate aminotransferase; ALT, alanine aminotransferase; ALB, albumin; PT, prothrombin time; NH₃, ammonia; HbA1c, hemoglobin A1c; HBV, hepatitis B virus; HCV, hepatitis C virus; ALD, alcohol-associated liver disease; MASLD, metabolic dysfunction-associated steatotic liver disease; LC, liver cirrhosis; HCC, hepatocellular carcinoma; ECW/TBW, extracellular water-to-total body water ratio; SMI, skeletal muscle mass index; KEF, knee extension force; SLST, single-leg stance test; DM, diabetes mellitus; HT, hypertension; DL, dyslipidemia; AWGS, Asian Working Group for Sarcopenia; JSH, Japanese Society of Hepatology.

**Table 2. Comparison of background characteristics between patients with and without DM among those with CLD (N = 152).**

| Characteristics | Patients with CLD (N = 152) | | p-value | Effect size |
|---|---|---|---|---|
| | Non-DM (n = 34) | DM (n = 118) | | |
| Age (years) | 63 (56–71) | 68 (63–73) | 0.005** | 0.23 (r) |
| Sex (male/female) | 23 (68)/ 11 (32) | 65 (55)/ 53 (45) | 0.238 | 0.11 (V) |
| Height (cm) | 163.6 (154.2–169.1) | 159.9 (153.6–166.2) | 0.188 | 0.11 (r) |
| Weight (kg) | 67.1 (62.3–86.9) | 63.0 (55.5–72.9) | 0.023* | 0.18 (r) |
| BMI (kg/m$^2$) | 27.1 (24.3–30.6) | 24.7 (22.0–28.8) | 0.025* | 0.18 (r) |
| Complications | | | | |
| HT | 16 (47) | 64 (54) | 0.559 | 0.06 (V) |
| DL | 8 (24) | 30 (25) | 1.000 | 0.02 (V) |
| Blood data | | | | |
| HGB (g/dL) | 13.9 (13.0–15.0) | 13.5 (12.0–14.8) | 0.145 | 0.12 (r) |
| PLT (10$^4$/μL) | 16.2 (11.1–22.2) | 15.9 (10.7–20.1) | 0.823 | 0.02 (r) |
| T-BIL (mg/dL) | 0.9 (0.5–1.3) | 0.9 (0.7–1.2) | 0.508 | 0.05 (r) |
| AST (IU/L) | 42 (28–61) | 29 (22–46) | 0.004** | 0.23 (r) |
| ALT (IU/L) | 42 (27–75) | 26 (18–47) | 0.001** | 0.27 (r) |
| ALB (g/dL) | 4.2 (3.7–4.5) | 4.3 (3.7–4.5) | 0.804 | 0.02 (r) |
| PT (%) | 84 (67–96) | 89 (77–98) | 0.301 | 0.08 (r) |
| NH$_3$ (μmol/L) | 39 (32–61) | 35 (27–49) | 0.084 | 0.14 (r) |
| HbA1c (%) | 5.8 (5.4–6.1) | 7.2 (6.6–8.0) | <0.001** | 0.59 (r) |
| Etiology (ALD/HBV/HCV/MASLD/other) | 3 (9)/ 4 (12)/ 14 (41)/ 9 (26)/ 2 (6) | 14 (12)/ 14 (12)/ 44 (37)/ 40 (34)/ 6 (5) | 0.633 | 0.13 (V) |
| LC | 2 (6) | 12 (10) | 0.737 | 0.06 (V) |
| HCC | 6 (18) | 26 (22) | 0.642 | 0.04 (V) |
| Encephalopathy | 0 (0) | 1 (1) | 1.000 | 0.04 (V) |
| Ascites | 2 (6) | 10 (8) | 1.000 | 0.04 (V) |
| Child–Pugh score | 5 (5–6) | 5 (5–6) | 0.862 | 0.01 (r) |
| Uncompensated cirrhosis; Child–Pugh score ≥ 7 | 7 (21) | 17 (14) | 0.425 | 0.07 (V) |
| Child–Pugh classification (A/B/C) | 27 (79)/ 5 (15)/ 2 (6) | 100 (85)/ 13 (11)/ 5 (4) | 0.760 | 0.06 (V) |

Continuous variables are presented as median (interquartile range). Categorical variables are presented as numbers (percentages). Effect sizes are interpreted as follows: r and Cramer's V (0.1–0.3 = small, 0.3–0.5 = medium, and ≥ 0.5 = large). *p < 0.05, **p < 0.01. CLD, chronic liver disease; DM, diabetes mellitus; BMI, body mass index; HT, hypertension; DL, dyslipidemia; HGB, hemoglobin; PLT, platelet; T-BIL, total bilirubin; AST, aspartate aminotransferase; ALT, alanine aminotransferase; ALB, albumin; PT, prothrombin time; NH$_3$, ammonia; HbA1c, hemoglobin A1c; ALD, alcohol-associated liver disease; HBV, hepatitis B virus; HCV, hepatitis C virus; MASLD, metabolic dysfunction-associated steatotic liver disease; LC, liver cirrhosis; HCC, hepatocellular carcinoma.

NPV, 91.7% (Fig 2d). Furthermore, the AUC for the combination of age, BMI, ECW/TBW, and KEF/weight was 0.795 (p < 0.001), with a sensitivity of 60.9%; specificity, 87.1%; PPV, 53.8%; and NPV of 90.0% (Fig 2e).

## Discussion

This study retrospectively investigated the association between DM and balance function in patients with CLD. To the best of our knowledge, this is the first study to evaluate SLST as an indicator of balance in this patient population. A comparison between patients with and without DM revealed a significant difference in the proportion of those with impaired SLST, defined as < 5 s. Furthermore, impaired SLST was significantly associated with age, BMI, and ECW/TBW. Among these, ECW/TBW demonstrated a cutoff value of 0.393, with a sensitivity of 73.9% and specificity of 68.8%.

**Table 3. Comparison of body composition and physical function between patients with and without DM in CLD (N = 152).**

| Characteristics | Patients with CLD (N = 152) | | | |
| --- | --- | --- | --- | --- |
| | Non-DM (n = 34) | DM (n = 118) | p-value | Effect size |
| ECW/TBW | 0.388 (0.380–0.395) | 0.391 (0.384–0.395) | 0.195 | 0.11 (r) |
| SMI (kg/m²) | 7.40 (6.48–9.14) | 6.97 (6.35–7.59) | 0.066 | 0.15 (r) |
| Male < 7.0, Female < 5.4 | 7 (21) | 34 (29) | 0.388 | 0.08 (V) |
| ASM/weight (%) | 27.1 (25.0–30.7) | 27.6 (23.8–31.5) | 0.767 | 0.02 (r) |
| ASM/BMI (kg/kg/m²) | 0.72 (0.59–0.83) | 0.70 (0.56–0.85) | 0.905 | 0.01 (r) |
| Upper limb muscle mass (kg/m²) | 2.04 ± 0.52 | 1.85 ± 0.39 | 0.029* | 0.43 (d) |
| KEF/weight (kgf/kg) | 0.57 ± 0.17 | 0.50 ± 0.15 | 0.011* | 0.50 (d) |
| SLST (s), Mean (Both legs) | 49.3 (22.9–60.0) | 23.9 (7.0–52.8) | 0.003** | 0.24 (r) |
| < 5 | 0 (0) | 23 (20) | 0.002** | 0.23 (V) |
| SLST (s), Max (Either leg) | 60 (28.4–60) | 30.0 (8.5–60) | <0.001** | 0.27 (r) |
| < 5 | 0 (0) | 16 (14) | 0.023* | 0.19 (V) |
| Grip strength (kg) | 28.8 (21.9–38.3) | 25.3 (19.4–32.5) | 0.031* | 0.18 (r) |
| Male < 28, Female < 18 | 8 (24) | 40 (34) | 0.299 | 0.09 (V) |
| Gait speed (m/s) | 1.15 ± 0.27 | 1.19 ± 0.24 | 0.331 | 0.19 (d) |
| < 1.0 | 8 (24) | 33 (28) | 0.825 | 0.04 (V) |
| Sarcopenia by 2019 AWGS | 5 (15) | 18 (15) | 1.000 | 0.01 (V) |
| Sarcopenia by JSH, 2nd edition | 4 (12) | 17 (15) | 0.786 | 0.03 (V) |

Continuous variables are presented as mean ± standard deviation or median (interquartile range), as appropriate. Categorical variables are presented as numbers (percentages). Effect sizes are interpreted as follows: Cohen's d (0.2–0.5 = small, 0.5–0.8 = medium, and ≥ 0.8 = large); r and Cramer's V (0.1–0.3 = small, 0.3–0.5 = medium, and ≥ 0.5 = large). *p < 0.05, **p < 0.01. DM, diabetes mellitus; CLD, chronic liver disease; ECW/TBW, extracellular water-to-total body water ratio; SMI, skeletal muscle mass index; ASM, appendicular skeletal muscle mass; BMI, body mass index; KEF, knee extension force; SLST, single-leg stance test; AWGS, Asian Working Group for Sarcopenia; JSH, Japanese Society of Hepatology.

**Table 4. Logistic regression analysis for impaired SLST (< 5 s) among patients with CLD and DM (n = 118).**

| Variables | Univariate analysis | | | Multivariate analysis | | | |
| --- | --- | --- | --- | --- | --- | --- | --- |
| | OR | 95% CI | p-value | OR | 95% CI | p-value | VIF |
| Age (years) | 1.070 | 1.013–1.139 | 0.014* | 1.089 | 1.020–1.177 | 0.009* | 1.277 |
| BMI (kg/m²) | 1.095 | 1.007–1.194 | 0.035* | 1.176 | 1.045–1.343 | 0.006* | 1.305 |
| NH₃ (µmol/L) | 0.998 | 0.974–1.020 | 0.880 | | | | |
| HbA1c (%) | 1.160 | 0.859–1.545 | 0.318 | | | | |
| Child–Pugh score | 1.104 | 0.785–1.479 | 0.539 | | | | |
| ECW/TBW × 1000 | 1.077 | 1.028–1.140 | 0.001* | 1.065 | 1.003–1.138 | 0.039* | 1.792 |
| SMI (kg/m²) | 1.039 | 0.708–1.495 | 0.837 | | | | |
| KEF/weight (kgf/kg) × 100 | 0.951 | 0.917–0.983 | 0.003* | 0.978 | 0.938–1.020 | 0.299 | 1.795 |

*p < 0.05. SLST, single-leg stance test; CLD chronic liver disease; DM, diabetes mellitus; OR, odds ratio; CI, confidence interval; VIF, variance inflation factor; BMI, body mass index; NH₃, ammonia; HbA1c, hemoglobin A1c; ECW/TBW, extracellular water-to-total body water ratio; SMI, skeletal muscle mass index; KEF, knee extension force.

In patients with CLD who had comorbid DM, the proportion of individuals with impaired SLST was higher than that in those without DM. The prevalence of DM among patients with CLD is well documented and varies depending on the underlying etiology. For example, in MASLD, approximately 56% of patients had type 2 DM [3]. Furthermore, 83% of patients with CLD referred from hepatology to rehabilitation departments had comorbid DM [4]. Consistent with these

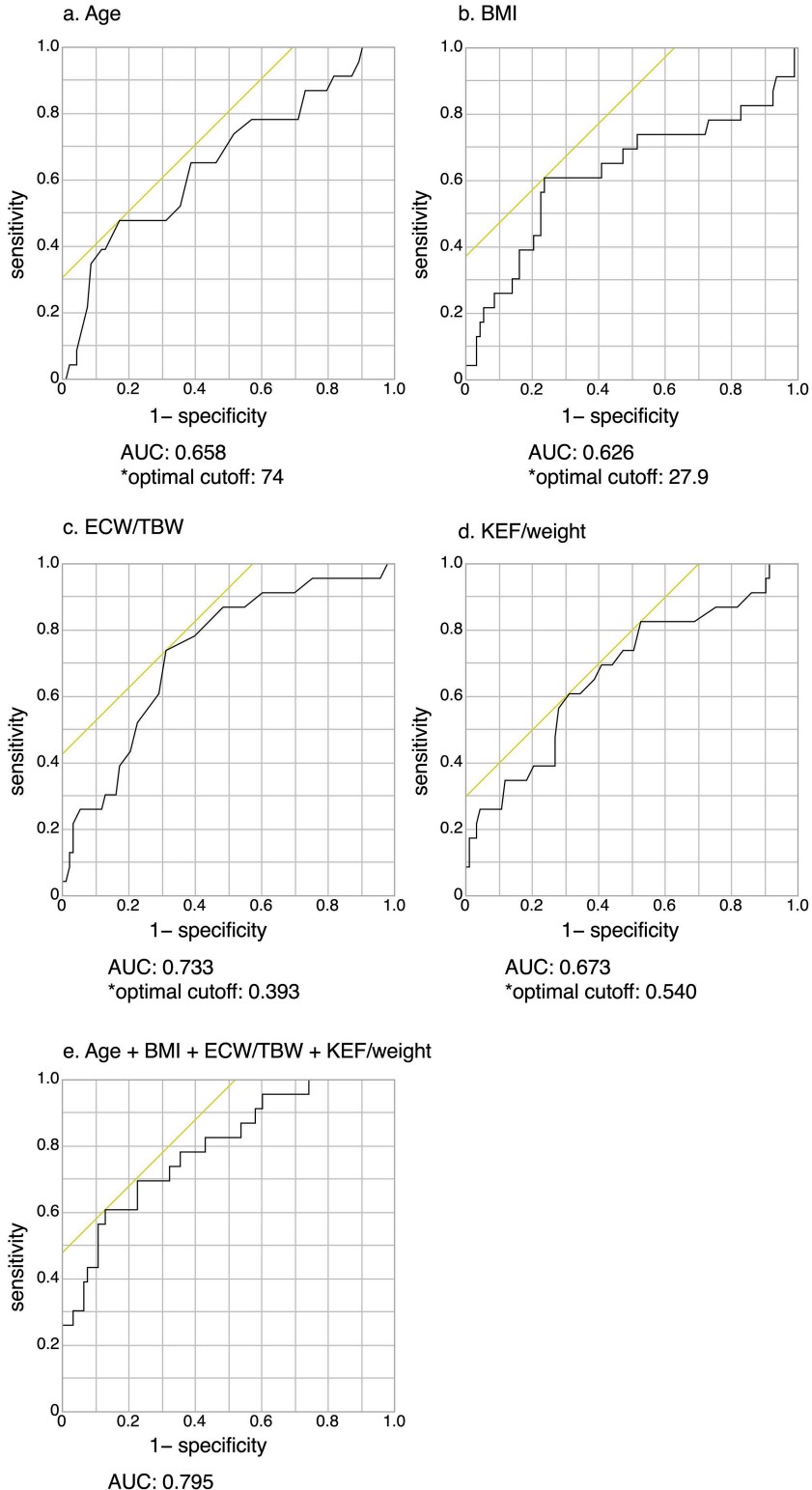

**Fig 2. ROC curves for predicting SLST duration<5s in patients with CLD and DM (n=118), using age, BMI, ECW/TBW, and KEF/weight as predictors.** ROC curves for (a) age, (b) BMI, (c) ECW/TBW, (d) KEF/weight, and the combination of age, BMI, ECW/TBW, and KEF/weight using SLST<5s as the dependent variable. The results showed the following: age: AUC=0.658, p=0.014, cutoff=74 years, sensitivity=47.8%, specificity=82.8%,

PPV = 40.7%, NPV = 86.5%; BMI: AUC = 0.626, p = 0.035, cutoff = 27.9 kg/m², sensitivity = 60.9%, specificity = 76.3%, PPV = 38.9%, NPV = 88.8%; ECW/TBW: AUC = 0.733, p = 0.001, cutoff = 0.393, sensitivity = 73.9%, specificity = 68.8%, PPV = 37.0%, NPV = 91.4%; KEF/weight: AUC = 0.673, p = 0.003, cutoff = 0.54 kgf/kg, sensitivity = 82.6%, specificity = 47.3%, PPV = 27.9%, NPV = 91.7%; and, the combination of age, BMI, ECW/TBW, and KEF/weight: AUC = 0.795, p < 0.001, sensitivity = 60.9%, specificity = 87.1%, PPV = 53.8%, NPV = 90.0%. ROC, receiver operating characteristic; SLST, single-leg stance test; CLD, chronic liver disease; DM, diabetes mellitus; BMI, body mass index; ECW/TBW, extracellular water-to-total body water ratio; KEF, knee extension force; AUC, area under the curve; PPV, positive predictive value; NPV, negative predictive value.

findings, 78% of the patients with CLD in this study had DM. The SLST is an established tool for predicting locomotive syndrome [34] and long-term fall risk over several years [32,33,41]. The present findings highlight the necessity of rehabilitation therapy aimed at fall prevention in patients with CLD and comorbid DM.

In patients with CLD and comorbid DM, age, BMI, and ECW/TBW were significantly associated with impaired SLST. Previous studies have demonstrated that aging is associated with impaired SLST, with sensory-function deterioration being one of the key contributing factors [42]. In addition to sensory decline, aging also leads to reductions in muscle strength, joint range of motion, and cognitive function, all of which are essential for maintaining balance in a single-leg stance. Although the DM group had a lower BMI than the non-DM group, higher BMI was identified as an independent risk factor within the DM cohort. Our findings regarding BMI are consistent with previous studies that have reported an association between obesity and impaired SLST [31,43,44]. In individuals with obesity, increased plantar pressure and contact area may reduce the quality and quantity of sensory information from plantar mechanoreceptors, leading to a decreased sensitivity of these receptors [44]. Weight loss has been reported to be effective in improving postural control [45], and exercise therapy may improve not only the underlying pathophysiology but also balance capability in patients with CLD and comorbid DM. In this study, no significant association was observed between $NH_3$ levels and SLST in the present study. Although hyperammonemia is known to impair motor coordination by enhancing GABAergic neurotransmission [46], most participants in our study did not have advanced cirrhosis, markedly elevated $NH_3$ levels, or overt hepatic encephalopathy, which may explain the lack of association observed.

Given that lower extremity muscle strength is more closely associated with walking ability than grip strength [47], we conducted an ROC analysis on a multivariable model incorporating age, BMI, ECW/TBW, and KEF/weight in relation to impaired SLST. Although the four-variable model achieved an AUC of 0.795, ECW/TBW alone also demonstrated acceptable predictive performance. Therefore, considering its high NPV (91.4%), an ECW/TBW cutoff of 0.393 appears well suited as an exclusion marker for impaired SLST. ECW/TBW increases as liver fibrosis progresses and has been reported as a marker of liver dysfunction and prognosis in patients with cirrhosis [48]. Moreover, ECW/TBW gradually increases in the pre-cirrhotic stage [48]. A threshold of ECW/TBW ≥ 0.400 is considered indicative of a moderate-to-severe overhydrated state [49]. Additionally, edema negatively affects physical function, including gait ability [50,51]. Similarly, in our study, increased ECW/TBW was associated with impaired SLST, suggesting that lower limb edema may contribute to reduced balance. Furthermore, ECW/TBW of 0.393 was identified as a potential predictor for ruling out impaired SLST in patients with CLD and comorbid DM. Moreover, DM is associated with fluid retention in patients with CLD [2,7]. Notably, DM has been shown to be independently associated with ascites development in patients with cirrhosis, irrespective of the model used for end-stage liver disease scores [7]. The underlying mechanisms of DM-associated fluid retention in CLD remain unclear; however, alterations in hepatic and renal microcirculation have been implicated [2]. From a rehabilitation perspective, even in cases where ECW/TBW does not exceed 0.400, values of 0.390–0.400 may represent a "subclinical" fluid retention state. It may be necessary to reconsider mild overhydration as an important factor in ruling out the risk of falls in patients with CLD and comorbid DM.

The association between increased ECW/TBW and impaired SLST in patients with CLD and comorbid DM likely involves multiple pathophysiological mechanisms. In liver cirrhosis, portal hypertension and lymphatic dysfunction lead to interstitial fluid accumulation and lower extremity edema [5,52], which may impair muscle strength and proprioception, thereby diminishing balance [53,54]. Furthermore, insulin resistance—a hallmark of both CLD and DM—may activate

proteasome and autophagy pathways via impaired signaling, potentially leading to muscle wasting and weakness [55,56]. Additionally, elevated serum myostatin levels in progressive CLD may suppress muscle protein synthesis [57,58]. Collectively, although the exact pathways remain unclear, changes in interstitial fluid dynamics, insulin resistance-induced signaling impairment, and disrupted liver–muscle crosstalk are potential factors linking increased ECW/TBW with impaired SLST.

This study has several limitations. First, due to its cross-sectional design, causal relationships cannot be established. Longitudinal prospective studies are required to further elucidate these associations. Second, blood tests were performed at admission, whereas body composition and physical function were assessed within three days after admission. Changes in clinical status during this interval, including alterations in fluid balance, may have introduced measurement bias, particularly for variables such as ECW/TBW that are sensitive to short-term fluctuations. Third, distinguishing between type 2 DM and hepatogenous DM in patients with CLD remains difficult, as noted in previous studies involving similar populations. Fourth, a major limitation of this study is the lack of assessment for diabetic peripheral neuropathy. As neuropathy is a primary mechanism for balance impairment in DM, it represents a significant unmeasured confounder. Therefore, we cannot exclude the possibility that the observed association between DM and impaired SLST is mediated, at least in part, by neuropathy rather than fluid distribution alone. Fifth, detailed information regarding the duration of DM, specific medications, and history of falls was not available. Sixth, the small sample sizes in the HBV, ALD, and other etiology groups limited our ability to detect significant differences between subgroups. Seventh, the limited variability in $NH_3$ levels among our participants, reflecting predominantly mild CLD, may have obscured any association with SLST results. Eighth, our study population had a remarkably high prevalence of DM (78%) and predominantly comprised patients with compensated liver disease (83% Child-Pugh A). This high selectivity may limit the generalizability of our findings to the broader CLD population, and the possibility of selection bias cannot be excluded. Ninth, the non-DM group had a relatively small sample size (n = 34) with a 0% incidence of impaired SLST. This lack of events in the control group may lead to unstable estimates in the comparative analysis, which should be considered when interpreting the results. Finally, the ECW/TBW cutoff value of 0.393 was derived from the present dataset, and external validation is needed prior to clinical application. Future prospective studies should aim to collect more detailed data on these variables to validate and expand upon our findings.

## Conclusions

DM was associated with a higher prevalence of impaired SLST in patients with CLD. Age, BMI, and ECW/TBW were identified as significant factors associated with impaired SLST. These findings suggest that SLST may be a useful tool for routine assessments of balance in patients with CLD and comorbid DM. Exercise therapy aimed at weight reduction may also contribute to improvements in balance. Furthermore, an ECW/TBW of 0.393 was identified as a potential cutoff for predicting impaired SLST. From a fall-prevention perspective, even mild overhydration (ECW/TBW 0.390–0.400) may warrant attention in patients with CLD and comorbid DM.

However, these findings should be interpreted cautiously given the cross-sectional design of the study, the lack of objective assessment of diabetic peripheral neuropathy, and potential measurement bias related to differences in assessment timing. Further prospective studies are required to confirm these associations and clarify their clinical implications.

## Supporting information

**S1 Table. Characteristics of the patients included in this study (N = 152).**
(DOCX)

**S2 Table. Comparison of background characteristics between patients with and without DM among those with CLD (N = 152).**
(DOCX)

## Acknowledgments

We would like to thank Editage (www.editage.jp) for English language editing.

## Author contributions

**Conceptualization:** Kenichi Fudeyasu, Toshihiro Kawae, Takuo Nomura, Yuki Nakashima, Daisuke Iwaki.

**Data curation:** Kenichi Fudeyasu, Makoto Asaeda, Yuki Nakashima, Daisuke Iwaki.

**Formal analysis:** Kenichi Fudeyasu, Toshihiro Kawae, Takuo Nomura, Yuki Nakashima, Daisuke Iwaki, Haruya Ohno, Eisuke Murakami.

**Funding acquisition:** Kenichi Fudeyasu, Makoto Asaeda, Yukio Mikami.

**Investigation:** Kenichi Fudeyasu, Toshihiro Kawae, Yuki Nakashima.

**Methodology:** Kenichi Fudeyasu, Toshihiro Kawae, Takuo Nomura.

**Project administration:** Kenichi Fudeyasu, Yukio Mikami.

**Resources:** Kenichi Fudeyasu, Eisuke Murakami, Shiro Oka, Yukio Mikami.

**Software:** Kenichi Fudeyasu.

**Supervision:** Makoto Asaeda, Shiro Oka, Yukio Mikami.

**Validation:** Yuki Nakashima, Haruya Ohno.

**Visualization:** Kenichi Fudeyasu, Yuki Nakashima.

**Writing – original draft:** Kenichi Fudeyasu, Yuki Nakashima, Haruya Ohno, Eisuke Murakami.

**Writing – review & editing:** Kenichi Fudeyasu, Makoto Asaeda, Toshihiro Kawae, Takuo Nomura, Yuki Nakashima, Daisuke Iwaki, Kouki Fukuhara, Takeya Araki, Haruya Ohno, Eisuke Murakami, Shiro Oka, Yukio Mikami.

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
