## [Decision Letter · Decision Letter 0]

21 Apr 2025

Dear Dr. Asaeda,

Thank you for submitting your manuscript to PLOS ONE. After careful consideration, we feel that it has merit but does not fully meet PLOS ONE’s publication criteria as it currently stands. Therefore, we invite you to submit a revised version of the manuscript that addresses the points raised during the review process.

Dear Authors,

Thank you for your submission to PLOS ONE. We have now received and reviewed two detailed and constructive peer reviews. Based on the feedback, the manuscript demonstrates scientific merit and clinical relevance; however, significant revisions are needed before it can be considered for publication.

In preparing your revised manuscript, please address the following key points in addition to the specific reviewer comments:

**Clarify and refine your hypothesis and objectives** to more precisely reflect your focus on balance impairment (i.e., SLST) rather than general physical function.**Revise and standardize methodological descriptions** , including the FIB-4 formula, consent procedure, variable definitions, and statistical approach (e.g., normality testing, multicollinearity diagnostics, use of exact logistic regression where appropriate).**Expand your discussion** to contextualize the ECW/TBW findings. Clearly note its current status as a data-derived marker needing external validation, and consider integrating relevant mechanistic pathways.**Ensure consistency** in terminology (e.g., always use “diabetes mellitus (DM)” after first mention), abbreviation use, and statistical formatting (e.g., p-values).

We look forward to receiving your revised manuscript.

Kind regards,

Bisher Sawaf

Academic Editor

PLOS ONE

“This work was supported by a grant (H30-A59) from the Japanese Society of Physical Therapy and Japan Society for the Promotion of Science KAKENHI Grant Number JP19K19831.”

3. In this instance it seems there may be acceptable restrictions in place that prevent the public sharing of your minimal data. However, in line with our goal of ensuring long-term data availability to all interested researchers, PLOS’ Data Policy states that authors cannot be the sole named individuals responsible for ensuring data access (http://journals.plos.org/plosone/s/data-availability#loc-acceptable-data-sharing-methods).

5. Please amend either the title on the online submission form (via Edit Submission) or the title in the manuscript so that they are identical.

Reviewers' comments:

Reviewer's Responses to Questions

**Comments to the Author**

1. Is the manuscript technically sound, and do the data support the conclusions?

Reviewer #1: Yes

Reviewer #2: Yes

2. Has the statistical analysis been performed appropriately and rigorously?

Reviewer #1: Yes

Reviewer #2: Yes

3. Have the authors made all data underlying the findings in their manuscript fully available?

Reviewer #1: Yes

Reviewer #2: Yes

4. Is the manuscript presented in an intelligible fashion and written in standard English?

Reviewer #1: Yes

Reviewer #2: Yes

Reviewer #1: Thank you to the authors for your thoughtful and well-organized manuscript.

Introduction (Lines 48–72)

Line 67: Reference specific studies showing increased fall risk due to diabetic neuropathy or proprioceptive impairment.

Line 71: Replace “poorer physical function” with “reduced physical function” for clarity and academic tone.

Line 72: The objective could be rephrased more precisely: “to examine whether comorbid diabetes mellitus is associated with impaired balance, as measured by single-leg standing time, in patients with chronic liver disease.”

Methods (Lines 74–192)

Line 84: Provide justification for the opt-out consent model. Mention whether this aligns with national research guidelines for retrospective studies in Japan.

Lines 96–98: There is repetition in the list of blood markers (e.g., AST, ALT, ALB) — revise to eliminate redundancy.

Line 109: The FIB-4 formula includes a square root of ALT, which is not conventional. Please clarify with a reference or correct if inaccurate.

Line 126: Specify whether SLST measurements had inter-rater or intra-rater reliability assessments.

Lines 170–173: Confirm whether normality of distribution was assessed before choosing statistical tests like t-tests or Mann-Whitney U. If so, report those results explicitly.

Results (Lines 194–295)

Line 199: Consider merging etiologies with low sample size (e.g., HBV, ALD, others) or acknowledging the limited power for these subgroups.

Line 243: A 0% vs. 20% SLST impairment rate is striking — discuss whether key confounders (e.g., diabetic neuropathy, medication use, or fall history) were considered or controlled.

Line 282: The predictive value of ECW/TBW (AUC = 0.733, PPV = 37%) is moderate. Discuss whether combining this marker with other functional metrics (e.g., grip strength) could improve diagnostic accuracy.

Discussion (Lines 297–366)

Line 304: Reaffirm how this is the first study (to your knowledge) evaluating SLST in CLD populations — cite any comparable work or clarify novelty.

Line 330: Address the low variance in ammonia levels (NH₃), which may explain the lack of association with SLST. Suggest this as a sample bias or a limitation of the study cohort.

Line 337: Include references to prior research connecting mild overhydration or edema to proprioceptive or postural instability to support the ECW/TBW mechanism.

Line 349: Recommend specific design elements for future research, e.g., prospective follow-up with diabetes duration, neuropathy scores, and functional falls data.

General Suggestions

Consistency: Standardize all uses of “diabetes” to “diabetes mellitus (DM)” after first mention for clarity.

Abbreviations: Suggest adding a footnote or separate table listing abbreviations like ECW/TBW, ASM/BMI, KEF, etc.

Statistics: Ensure consistent p-value formatting (e.g., p = 0.002). If exact p-values are unavailable, use p < 0.001 rather than zero.

Reviewer #2: Introduction:

- The hypothesis is stated as a general decline in “physical function” among DM patients with CLD. However, the actual analysis focuses narrowly on balance function as assessed by SLST. I suggest reframing the hypothesis to specifically address postural stability or balance-related impairment, aligning better with the operationalized outcome.

Methods:

- The FIB-4 formula includes a square root of ALT, which deviates from the canonical form (without transformation). This must be clarified, as it could affect the fibrosis score interpretation across studies.

- SLST methodology uses the mean of both legs, but no indication is given about distributional skew or floor effects. Given that 20% of DM patients had SLST <5 seconds and non-DM had 0%, consider whether a dichotomous transformation (e.g., <5s vs ≥5s) might yield more interpretable or clinically meaningful stratification for this population.

- It is unclear whether leg dominance was recorded. While cited literature claims no performance difference, individual patient-level variability may still affect balance metrics, particularly in neuromuscular-compromised cohorts.

- There is no mention of multicollinearity diagnostics before entering variables into multivariate logistic regression. BMI and ECW/TBW may be collinear. Reporting variance inflation factors (VIFs) or tolerance values is recommended.

Results:

- The finding of 0% SLST <5s in non-DM patients vs. 20% in DM patients is striking but statistically vulnerable due to zero-cell counts. Consider whether exact logistic regression would be more appropriate than asymptotic methods here.

- The AUC for ECW/TBW is 0.733 with relatively low PPV (37%) but high NPV (91.4%). This indicates that ECW/TBW may serve better as a rule-out marker than a rule-in diagnostic. The authors should address this explicitly in interpretation.

Discussion (Lines 297–366)

- The discussion of ECW/TBW as "subclinical edema" is compelling. However, the threshold of 0.393 is data-derived from this study and requires external validation before being used in clinical settings. This caveat should be emphasized.

- There’s an opportunity here to draw connections between interstitial fluid dynamics, insulin resistance, and muscle signaling pathways, all of which are altered in CLD-DM. These links could provide a more mechanistic narrative for the ECW/TBW-SLST association.

**Do you want your identity to be public for this peer review?** For information about this choice, including consent withdrawal, please see our Privacy Policy

Reviewer #1: No

Reviewer #2: No

---

## [Author Response · Author response to Decision Letter 1]

3 Jun 2025

May 28, 2025

Dr. Bisher Sawaf

Academic Editor

PLOS ONE

Dear Editor,

Thank you for your consideration of our manuscript previously titled “Chronic liver disease complicated by diabetes mellitus is associated with impaired single-leg standing in patients: a cross-sectional study” (Manuscript ID: [PONE-D-25-15349], EMID: 7be85993d2e2226f).

We have carefully revised the manuscript in accordance with the editors’ and reviewers’ suggestions. A point-by-point response to each comment is provided below. All changes in the revised manuscript are highlighted in red font. We sincerely hope that our responses and revisions have satisfactorily addressed all concerns, and that the manuscript is now suitable for publication in PLOS ONE.

Please note that we have updated the manuscript title to replace “Chronic liver disease complicated by diabetes mellitus is associated with impaired single-leg standing in patients: a cross-sectional study” with the more standard and academically appropriate title “Association between diabetes mellitus and impaired single-leg stance in patients with chronic liver disease: a cross-sectional study.”

Additionally, we would like to request an update to the second affiliation of the first author, Kenichi Fudeyasu. The corrected affiliation is:

2 Department of Rehabilitation Medicine, Osaka Medical and Pharmaceutical University Hospital, Osaka, Japan.

Please ensure this correction is reflected in the author list and manuscript metadata upon acceptance.

We greatly appreciate the opportunity to revise our work and look forward to your response.

Best regards,

Makoto Asaeda

Department of Rehabilitation Medicine, Hiroshima University Hospital

1-2-3 Kasumi, Minami-ku, Hiroshima 734-8551, Japan TEL: +81-82-257-5566

E-mail: asaedam@hiroshima-u.ac.jp

To the Editor’s comments

In preparing your revised manuscript, please address the following key points in addition to the specific reviewer comments:

- Clarify and refine your hypothesis and objectives to more precisely reflect your focus on balance impairment (i.e., SLST) rather than general physical function.

Response: Thank you very much for your insightful comment. In response to your suggestion, we have revised the description of the hypothesis and objective to better clarify the focus of our study, particularly highlighting the single-leg stance test (SLST) as the primary outcome. The revised text in the manuscript is as follows:

(Introduction, Page 5, Lines 88–91):

"We hypothesized that comorbid DM is associated with impaired balance in patients with CLD and that this decline would be evident in the single‑leg stance test (SLST) as a measurement of balance. Therefore, the aim of this study was to examine whether comorbid DM is associated with impaired SLST in patients with CLD."

- Revise and standardize methodological descriptions, including the FIB-4 formula, consent procedure, variable definitions, and statistical approach (e.g., normality testing, multicollinearity diagnostics, use of exact logistic regression where appropriate).

Response: Thank you very much for your important and constructive comments. In response, we have clarified several methodological aspects, including the formula for calculating the FIB-4 index, the informed consent process, variable definitions, and our statistical approach (e.g., normality testing, multicollinearity diagnostics, and use of exact logistic regression when appropriate). These revisions have been made in multiple sections of the manuscript and we have included a copy below for your reference:

(Methods, Page 5, Lines 103–112):

"The study was conducted in accordance with Declaration of Helsinki and the Ethical Guidelines for Medical and Health Research Involving Human Subjects enacted by the Ministry of Health, Labour and Welfare of Japan (https://www.mhlw.go.jp/content/001077424.pdf), and approved by the Ethical Review Committee of Hiroshima University Hospital (Permit No. E-583-1). All patients were verbally informed that their medical records and charts might be used for research purposes. Data were obtained during routine medical care and reviewed retrospectively. The need to obtain informed consent from the study participants was waived by the Ethical Review Committee of Hiroshima University Hospital due to the retrospective nature of the study. Patients who were eligible for this study had the opportunity to refuse to participate in the study by opting out."

(Methods, Page 7, Lines 134–138):

“Liver fibrosis was assessed using serum marker-based scoring systems, including the AST-to-platelet ratio index (APRI) and the fibrosis-4 (FIB-4) index [49–53]. The APRI and FIB-4 index were calculated using the following equations,

APRI=(AST[IU/L]/upper limit of normal AST[IU/L])/(PLT [〖10〗^9/L] )×100

FIB-4 index=(AST[IU/L]×age [years])/(PLT [〖10〗^9/L]×√(ALT [IU/L] ))

(Methods, Page 10, Lines 204–217):

“All continuous variables were first subjected to the Shapiro–Wilk test to assess normality. Variables that followed a normal distribution were presented as mean (± standard deviation) and were compared between groups using the Student’s t-test. Variables that deviated from normality were expressed as median (interquartile range) and compared using the Mann–Whitney U test. Categorical variables were presented as absolute numbers and percentages and compared using the Fisher’s exact test or chi-squared test, as appropriate.

An additional analysis was conducted using data from 118 patients with CLD who had comorbid DM, to identify factors associated with impaired SLST in this subgroup. Logistic regression analysis was performed with impaired SLST (< 5 s) as the dependent variable, using both univariate and multivariate models, and ORs with 95% CIs were reported. Following previous studies, KEF/weight values were multiplied by 100 and ECW/TBW values by 1000 to facilitate interpretation [66]. Multicollinearity was assessed by calculating variance inflation factors (VIFs), with values ≥ 2 considered indicative of collinearity.”

- Expand your discussion to contextualize the ECW/TBW findings. Clearly note its current status as a data-derived marker needing external validation, and consider integrating relevant mechanistic pathways.

Response: Thank you for these valuable suggestions. In response, we have expanded the discussion to better contextualize the significance of ECW/TBW. In particular, we have clarified that ECW/TBW is a data-derived marker that requires external validation, and we have incorporated relevant mechanistic pathways to support its interpretation. Specifically, we have made the following additions:

(Discussion, Page 24, Lines 395–419):

" The association between ECW/TBW and SLST in patients with CLD and comorbid DM may be explained by the interplay of multiple pathophysiological mechanisms. In liver cirrhosis, portal hypertension and lymphatic dysfunction lead to the accumulation of interstitial fluid [79, 80], resulting in pronounced lower extremity edema [81]. Such edema may impair muscle strength and proprioception, potentially leading to diminished balance capabilities [82-84]. Furthermore, CLD with comorbid DM is characterized by a significant resistance to insulin that has been linked to reductions in skeletal muscle mass and strength [85, 86]. In mice lacking insulin receptors (IR) and insulin-like growth factor-I receptors (IGF-IR) in skeletal muscles, protein catabolism is enhanced through activation of the proteasome and autophagy pathways, leading to muscle wasting and weakness. Notably, muscle atrophy is suppressed in mice deficient in Forkhead box O, indicating the critical role of IR/IGF-IR/Forkhead Box O signaling in maintaining muscle mass and strength [87]. In addition, mice lacking insulin signaling molecules such as phosphoinositide 3-kinase [88] and Akt [89] exhibit not only impaired glucose tolerance but also reductions in muscle mass and fiber diameter, further supporting the relationship between insulin resistance and muscle dysfunction. Myostatin, a myokine that strongly suppresses muscle protein synthesis, plays a key role in maintaining muscle homeostasis in accordance with anabolic myokines [90]. In patients with CLD, serum myostatin levels tend to increase as the disease progresses, potentially contributing to suppressed muscle protein synthesis [91, 92]. Taken together, these findings suggest that changes in interstitial fluid dynamics, insulin resistance-induced impairment of muscle signaling, and disruption of liver–muscle crosstalk act in concert to link increased ECW/TBW with impaired SLST. The relationship between decreased SLST and elevated ECW/TBW demonstrated in this study is not a mere correlation, but rather reflects a comprehensive pathophysiological process involving fluid imbalance, disrupted metabolic signaling, and changes in muscle quality."

(Discussion, Page 26, Lines 430–433):

“Finally, the ECW/TBW cutoff value of 0.393 was derived from the present dataset, and external validation is needed prior to clinical application. Future prospective studies should aim to collect more detailed data on these variables to confirm and expand upon our findings.”

- Ensure consistency in terminology (e.g., always use “diabetes mellitus (DM)” after first mention), abbreviation use, and statistical formatting (e.g., p-values).

Response: Thank you for your helpful comment. We have revised the manuscript to ensure consistency in terminology. Specifically, we used the full term “diabetes mellitus (DM)” at its first appearance and used the abbreviation “DM” thereafter throughout the text. Additionally, we standardized the presentation of p-values to either “p = 0.xxx” or “p < 0.001” as appropriate.

To the Journal requirements

Response: Thank you for your valuable comment. We have adjusted the manuscript formatting, including file naming, in accordance with the PLOS ONE style requirements.

“This work was supported by a grant (H30-A59) from the Japanese Society of Physical Therapy and Japan Society for the Promotion of Science KAKENHI Grant Number JP19K19831.”

Response: Thank you for your confirmation. We have included the following statement regarding the role of the funders in the cover letter:

(Response letter, Page 1)

3. In this instance it seems there may be acceptable restrictions in place that prevent the public sharing of your minimal data. However, in line with our goal of ensuring long-term data availability to all interested researchers, PLOS’ Data Policy states that authors cannot be the sole named individuals responsible for ensuring data access (http://journals.plos.org/plosone/s/data-availability#loc-acceptable-data-sharing-methods).

Response: Thank you for your comment. We confirm that data access will not be solely managed by the authors. We have arranged with the Hiroshima University Library to archive the dataset in the Hiroshima University Institutional Repository (https://ir.lib.hiroshima-u.ac.jp), and they have agreed to act as the non-author institutional contact for data access inquiries. Please direct any inquiries regarding the dataset to:

Hiroshima University Library

Email: tosho-soumu@office.hiroshima-u.ac.jp

Website: https://www.lib.hiroshima-u.ac.jp/

The library has committed to long-term preservation and public access to the dataset, ensuring that data remain available even if authors become unreachable.

Response: We confirm that the minimal dataset will be deposited in the Hiroshima University Institutional Repository and made freely and publicly accessible upon acceptance of the manuscript. The Hiroshima University Library has already agreed to this and the registration process is underway. We will update the Data Availability Statement in the manuscript as follows:

Data Availability:

“All relevant data will be made available without restriction in the Hiroshima University Institutional Repository (https://ir.lib.hiroshima-u.ac.jp) following acceptance. Data requests may also be directed to Hiroshima University Library (tosho-soumu@office.hiroshima-u.ac.jp), which will manage long-term data access.”

5. Please amend either the title on the online submission form (via Edit Submission) or the title in the manuscript so that they are identical.

Response: Thank you for your valuable comment. We have standardized the manuscript title in both the online submission system and within the manuscript itself as follows:

(Page 1, Lines 4–5)

“Association between diabetes mellitus and impaired single-leg stance in patients with chronic liver disease: a cross-sectional study”

To the Reviewer: 1

Introduction (Lines 48–72)

- Line 67: Reference specific studies showing increased fall risk due to diabetic neuropathy or proprioceptive impairment.

Response: Thank you very much for your valuable comment. In accordance with your suggestion, we have added several key references to support this point. The revised manuscript now states:

(Introduction, Page 4, Lines 82–86):

"Patients with DM also experience impaired balance due to factors such as peripheral neuropathy, hypoglycemia, and delayed reaction times [42-44]; the presence of DM alone increases the risk of falls [45]. Peripheral

---

## [Decision Letter · Decision Letter 1]

23 Dec 2025

Dear Dr. Asaeda,

Thank you for submitting your manuscript to PLOS ONE. After careful consideration, we feel that it has merit but does not fully meet PLOS ONE’s publication criteria as it currently stands. Therefore, we invite you to submit a revised version of the manuscript that addresses the points raised during the review process.

We look forward to receiving your revised manuscript.

Kind regards,

Rafael Oliveira

Academic Editor

PLOS One

Journal Requirements:

**Additional Editor Comments:**

Dear authors,

The authors improved their by following the comments made by reviewer in the first division. However, there are still margin for improvements.

Therefore, another round of revisions will be requested. Please considerer the comments made by reviewers and see the following specific comments for more details:

-in abstract, please add effect sizes when it's appropriate;

-L60-63 require at least one more citation or please rewrite the sentence;

-L87-91, please support your hypothesis with references;

-in materials and methods section, the order of evaluations and rest between them should be described;

-please add the type of the study;

-L140, please describe the time of the day for the assessment and what were the conditions applied before the assessment;

-L142, please add information about validity of the device;

-L206, effect sizes should be calculated and added for the appropriate analysis;

-Table 1, please check table format and size because it is too long (3 pages). In addition, please add units for all variables;

-for tables 2 and 3, please add effect sizes and their interpretation

Best regards

Reviewers' comments:

Reviewer's Responses to Questions

**Comments to the Author**

Reviewer #3: (No Response)

2. Is the manuscript technically sound, and do the data support the conclusions?

Reviewer #3: Yes

3. Has the statistical analysis been performed appropriately and rigorously?

Reviewer #3: Yes

4. Have the authors made all data underlying the findings in their manuscript fully available?

Reviewer #3: (No Response)

5. Is the manuscript presented in an intelligible fashion and written in standard English?

Reviewer #3: Yes

Reviewer #3: Manuscript Title: Association between diabetes mellitus and impaired single-leg stance in patients with chronic liver disease: a cross-sectional study

Journal: PLOS One

Authors: Makoto Asaeda et al.

Decision: Minor Revision

Rationale: The manuscript addresses a clinically important question and presents novel findings regarding ECW/TBW. However, some concerns need to be thoroughly addressed before the manuscript can be considered for publication.

Key Revisions Requested

Abstract

1. Line 45: "Patients with DM had a significantly higher prevalence of impaired SLST than those without DM (20% vs 0%, p = 0.002)." A 0% rate in the non-DM group is striking. While statistically significant, it is worth noting that the non-DM group was small (n = 34), and this result may be sensitive to sample size.

Introduction

1. Lines 87-91: The hypothesis and aim are clear. Well-stated.

Methods

1. Timing of Assessments (Lines 95-103, 118-121): The protocol where blood tests (admission) and physical function assessments (start of physiotherapy) occurred at different, unspecified times is a significant flaw. Clinical status, especially fluid balance (ECW/TBW), can change rapidly, introducing substantial measurement bias and weakening the core cross-sectional associations.

2. Unmeasured Confounding - Diabetic Neuropathy (Throughout): The most critical limitation is the absence of any assessment or adjustment for diabetic peripheral neuropathy. As a primary mechanism for balance impairment in DM, neuropathy is a profound confounder. The observed association between DM and impaired SLST may be entirely mediated by neuropathy, drastically altering the interpretation of the results.

3. Sample Generalizability (Lines 232-236): The cohort, with a remarkably high DM prevalence (78%) and predominantly compensated liver disease (83% Child-Pugh A), is highly selective. This limits the generalizability of the findings to the broader CLD population and should be explicitly acknowledged as a limitation.

Results

1. Lines 272-279: When presenting comparative results (e.g., KEF/weight, grip strength), ensure that the descriptive statistics (mean ± SD vs. median [IQR]) match the tests used (t-test vs. Mann-Whitney U) as reported in the tables.

Discussion

1. Lines 368-371: The discussion about the null finding for ammonia is reasonable given the population's characteristics.

2. Lines 392-394, 441-443: The concept of "subclinical edema" is interesting and a useful clinical takeaway. This is well-argued.

3. Lines 395-419: The pathophysiological discussion is comprehensive but somewhat speculative, especially the detailed molecular mechanisms (IR/IGF-IR/FoxO, myostatin), given that the study did not measure any of these parameters. This section could be shortened and framed more as potential explanatory hypotheses for future research rather than established explanations for the current findings.

Conclusion

1. The conclusions are generally supported by the data but should be tempered by acknowledging the key limitations, such as the cross-sectional design, lack of data on neuropathy, and the specific characteristics of the study cohort.

**Do you want your identity to be public for this peer review?** For information about this choice, including consent withdrawal, please see our Privacy Policy

Reviewer #3: No

---

## [Author Response · Author response to Decision Letter 2]

4 Feb 2026

February 4, 2026

Dr. Bisher Sawaf

Academic Editor

PLOS ONE

Dear Dr. Sawaf:

Thank you for your consideration of our revised manuscript. We are grateful to the reviewers for their insightful and constructive feedback. We have addressed all the concerns raised by the editor and Reviewer #3. A point-by-point response is provided below. In this revised manuscript, we have addressed all the reviewers' concerns, specifically regarding the standardizing of body composition assessment protocols, the clarification of statistical effect sizes, and the refinement of the study’s limitations. All changes in the revised manuscript are highlighted in red font.

Please ensure this correction is reflected in the author list and manuscript metadata upon acceptance.

We greatly appreciate the opportunity to revise our work and look forward to your response.

Best regards,

Makoto Asaeda

Department of Rehabilitation Medicine, Hiroshima University Hospital

1-2-3 Kasumi, Minami-ku, Hiroshima 734-8551, Japan TEL: +81-82-257-5566

E-mail: asaedam@hiroshima-u.ac.jp

---

## [Editor Report · Decision Letter 2]

8 Feb 2026

Dear Dr. <!--StartFragmentMakoto Asaeda<!--EndFragment,

Thank you for submitting your manuscript to PLOS ONE. After careful consideration, we feel that it has merit but does not fully meet PLOS ONE’s publication criteria as it currently stands. Therefore, we invite you to submit a revised version of the manuscript that addresses the points raised during the review process.

**ACADEMIC EDITOR:**

The authors improved their work, but there is still an issue that avoid my recommendation of acceptance. Thus, I'll require another round of revisions.

Specifically, please consider the following comments:

-L146-148, "To ensure data consistency, all measurements were conducted during the daytime and at least 2 hours after the last meal." Can you support this methodological choice with references, since this is not a standard procedure for body composition analysis? Moreover, were some procedures regarding food and water intake before the assessment?

-L220-221, for Cohen's D, R and V Cramer's, references to support the values of interpretations are needed. In addition, rather than specific values, authors should be present ranges for each interpretation, for better clarity;

-L438-439, in theses sentences there are two "sixth" limitations. The second time should be replaced by "seventh".

-96 references are too much for an original manuscript. This number should be reduced and not exceed 60.

Best regards

We look forward to receiving your revised manuscript.

Kind regards,

Rafael Oliveira

Academic Editor

PLOS One

Journal Requirements:

Additional Editor Comments (if provided):

Dear authors,

The authors improved their work, but there is still an issue that avoid my recommendation of acceptance. Thus, I'll require another round of revisions.

Specifically, please consider the following comments:

-L146-148, "To ensure data consistency, all measurements were conducted during the daytime and at least 2 hours after the last meal." Can you support this methodological choice with references, since this is not a standard procedure for body composition analysis? Moreover, were some procedures regarding food and water intake before the assessment?

-L220-221, for Cohen's D, R and V Cramer's, references to support the values of interpretations are needed. In addition, rather than specific values, authors should be present ranges for each interpretation, for better clarity;

-L438-439, in theses sentences there are two "sixth" limitations. The second time should be replaced by "seventh".

-96 references are too much for an original manuscript. This number should be reduced and not exceed 60.

Best regards

---

## [Author Response · Author response to Decision Letter 3]

1 Mar 2026

March 2, 2026

Dr. Rafael Oliveira

Academic Editor

PLOS ONE

Dear Dr. Oliveira,

Thank you for the further review of our manuscript, Manuscript ID: PONE-D-25-15349R2, Title: Association between diabetes mellitus and impaired single-leg stance in patients with chronic liver disease: a cross-sectional study. We appreciate your helpful comments, which have allowed us to refine our work even further. We have addressed all your suggestions in the revised manuscript. Our point-by-point responses are provided below.

Best regards,

Makoto Asaeda

Department of Rehabilitation Medicine, Hiroshima University Hospital

1-2-3 Kasumi, Minami-ku, Hiroshima 734-8551, Japan TEL: +81-82-257-5566

E-mail: asaedam@hiroshima-u.ac.jp

---

## [Editor Report · Decision Letter 3]

5 Mar 2026

Association between diabetes mellitus and impaired single-leg stance in patients with chronic liver disease: a cross-sectional study

PONE-D-25-15349R3

Dear Dr. Makoto Asaeda,

We’re pleased to inform you that your manuscript has been judged scientifically suitable for publication and will be formally accepted for publication once it meets all outstanding technical requirements.

Kind regards,

Rafael Oliveira

Academic Editor

PLOS One

Additional Editor Comments (optional):

Dear authors,

Thank you by following all suggestions. The manuscript was improved.

There is only one issue related with in text citations because when there is more more than one reference, they should be merged and the current version did not present them in that way.

Still, the manuscript can be accepted and the issue be fixed during the proofreading stage.

Congratulations!

Best regards